# Microbiota Co-Metabolism Alterations Precede Changes in the Host Metabolism in the Early Stages of Diet-Induced MASLD in Wistar Rats

**DOI:** 10.3390/ijms26031288

**Published:** 2025-02-02

**Authors:** María Martín-Grau, Pilar Casanova, Laura Moreno-Morcillo, José Manuel Morales, Vannina G. Marrachelli, Daniel Monleón

**Affiliations:** 1Departament de Patologia, Universitat de València, 46010 Valencia, Spain; pilar.casanova@uv.es (P.C.); laura.moreno-morcillo@uv.es (L.M.-M.); j.manuel.morales@uv.es (J.M.M.); 2INCLIVA Biomedical Research Institute, 46010 Valencia, Spain; vannina.gonzalez@uv.es; 3Departament de Fisiologia, Universitat de València, 46010 Valencia, Spain

**Keywords:** MASLD, metabolomics, NMR, faecal microbiota, sexual dimorphism

## Abstract

Metabolic-dysfunction-associated steatotic liver disease (MASLD) affects around 30% of the global population. The sexual dimorphism and gut microbiota play an important role in the early development of MASLD. The main objective of this research was to investigate metabolic changes during the early subclinical MASLD progression, for identifying the sequence of events and evaluating the impact of sexual dimorphism and the microbiota on the initial stages of MASLD development. Male and female Wistar rats 18 weeks old were randomly divided into different groups and fed a chow diet or a 45% high-fat diet for 21 weeks. Every three weeks, samples of serum, urine, and faeces were collected and studied by metabolomics. Furthermore, the liver was analysed at the endpoint. In addition, the gut microbiota was analysed from faecal samples over time using 16S rRNA gene-targeted group-specific primers. Our results revealed that three weeks on an HFD reduced the bacterial diversity in the faecal microbiota of Wistar rats, accompanied by changes in the faecal and urine metabolome. The HFD-induced alterations in microbiota-related co-metabolites in the liver, blood, urine, and faeces indicate a significant role of host–microbiota co-metabolism changes in the early stages of MASLD. In this study, we provide a comprehensive longitudinal analysis, detailing the sequence of events in the early development of MASLD. Our findings suggest that alterations in the gut microbiota diversity and co-metabolism occur before changes in host metabolism in the early onset of liver steatosis, a subclinical phase of MASLD.

## 1. Introduction

Metabolic-dysfunction-associated steatotic liver disease (MASLD) has emerged as a global health epidemic, affecting an estimated 30% of the world’s population [1]. MASLD encompasses a spectrum of stages, from simple steatosis, where patients remain asymptomatic, to severe conditions such as metabolic dysfunction-associated steatohepatitis (MASH), fibrosis, cirrhosis, and hepatocellular carcinoma (HCC). The prevalence and potential progression of MASLD to severe liver-related complications pose a significant public health burden [2].

High-fat diets, characterised by excessive saturated fatty acid (SFA) intake, are strongly linked to the development of MASLD [3]. Common in Western countries, these diets significantly influence gut microbiota composition [4]. Moreover, the gut-liver axis plays a pivotal role, as gut-derived endotoxins like lipopolysaccharides (LPS) and metabolites can reach the liver, triggering inflammation and metabolic disruptions [5]. These processes contribute to fat accumulation in the liver, which can progress to MASH and, in severe cases, fibrosis or cirrhosis [6]. Despite this, the specific role of host–microbiota co-metabolism in the early stages of MASLD remains poorly understood.

Causes precede effects. Longitudinal studies are invaluable for confirming that hypothetical causes precede effects, as they track the same subjects over an extended period. This approach allows researchers to observe changes and developments in real-time, establishing a clear temporal sequence between potential causes and outcomes. By consistently monitoring subjects, longitudinal studies can identify patterns and causal relationships that might be missed in cross-sectional studies, thereby providing robust evidence that supports or refutes hypothetical causation. The recent adoption of the term MASLD underscores its nature as a metabolic disease, positioning metabolomics as a tool to explore dynamic changes in metabolites during disease progression [7,8]. Metabolomics is an excellent technique for conducting longitudinal studies.

The primary objective of this research was to longitudinally investigate metabolic changes and microbiota alterations in a subclinical MASLD model for elucidating the sequence of events during early fatty liver progression in males and females, for identifying metabolites alterations with potential value as proxy markers of subclinical disease and providing time sequence support for previously suggested cause–effect relationships.

## 2. Results

### 2.1. Twenty-One Weeks of HFD Induced Subclinical Liver Alterations Not Reflected in Blood Parameters

Physiological and biochemical measurements are crucial in managing MASLD. We characterised the effects of a 45% HFD over 21 weeks in our model (Table 1). HFD animals consumed more energy daily, leading to increased body weight (BW), body mass index (BMI), and liver weight (LW) in HFD males by the experiment’s end. However, these differences were not statistically significant in HFD females. Blood cell counts showed differences in neutrophil levels only in HFD males. General biochemistry revealed a significant increase in alkaline phosphatase (ALP) in HFD males and a significant decrease in bile acids (BAs) in HFD animals. Despite 21 weeks of HFD, basal glycaemia values and insulin resistance did not significantly increase in both male and female Wistar rats (Appendix A). The only notable difference was a quicker glucose response in CTL females compared to males at t21 (Appendix A). Routine tests did not indicate MASLD after 21 weeks of HFD. However, histopathological assessment revealed early onset liver steatosis without evidence of fibrosis development (Figure 1A). ORO stain quantification showed greater positive areas in both HFD males and females compared to their controls, with HFD males exhibiting higher ORO-positive areas than HFD females (Figure 1B). This suggests that HFD males accumulated more fat within hepatocytes than HFD females. Histopathological studies remain key in diagnosing MASLD.

### 2.2. HFD-Induced Liver Metabolome Changes Were Sex-Specific

After confirming early-onset of liver steatosis, we performed metabolomic characterisation of the liver at the endpoint. The PLS-DA score plot separated the HFD and CTL groups, showing separation between males and females only in the HFD group (Figure 1C). In males, significant differences were observed in tau-methylhistidine, formate, alpha-glucose, trimethylamine N-oxide (TMAO), and inosine (Figure 1E). Formate and TMAO are related to host–microbiota co-metabolism [9,10]. In females, significant differences were only observed in polyunsaturated fatty acids (PUFAs) (Figure 1G). These results indicate sex-specific differences in liver metabolome response to HFD, with males showing greater differences than females. The lipid moieties profile (Appendix A) revealed that not all fats increased despite histological steatosis. In males, significant increases in SFAs, long-chain carbonyls (lcCOs), long-chain unsaturated fatty acids (lcUFAs), and total carbonyl groups (tCOs) were observed in HFD males compared to CTL males. Conversely, PUFAs and total unsaturated fatty acids (tUFAs) significantly decreased in HFD males. In HFD females, significant increases in SFAs and lcUFAs and decreases in PUFAs and tUFAs were observed. Males had higher amounts of SFAs, lcUFAs, and tCOs, while females had higher amounts of PUFAs and tUFAs.

### 2.3. Serum Metabolic Signature Changes Induced by 21 Weeks of HFD Are Minor, Except for Lipid Moieties

We investigated serum metabolome changes in male and female Wistar rats every three weeks. The PLS-DA score plot at t0 and t21 (Figure 2A) showed that while females remained mixed at t21, males could be separated into CTL and HFD groups. Twenty-one weeks of HFD did not generate a clear metabolic signature in serum (Figure 2B), although differences in N-acetyl compounds (Figure 2C) and PUFAs (Figure 2D) were observed from week 3 onwards. tUFAs, TMAO, and choline compounds did not show statistically significant differences over time but had relevant results. tUFA levels decreased only in HFD females (Figure 2E). TMAO levels decreased in HFD animals until t21 (Figure 2F). Choline compounds fluctuated over time but generally decreased in HFD animals (Figure 2G). The serum lipid moiety profiles at the endpoint (Appendix A) showed a significant increase in SFA in HFD males compared to CTL males, with significant decreases in lcUFAs, PUFAs, and tUFAs. In females, significant increases in tCO and decreases in tUFAs were observed in HFD females. Males had higher SFA levels, while females had higher lcCO, tCO, and PUFA levels. Comparing serum and liver lipid profiles at the endpoint, HFD males accumulated more detrimental fat in the liver, while HFD females had more detrimental fat in circulating blood. Both serum and liver of HFD animals showed decreased PUFAs and tUFAs, with a clearer pattern in males (Appendix A).

### 2.4. HFD Induces Early Sex-Dimorphic Changes in Urine Metabolome

We analysed the urine metabolome in male and female Wistar rats every three weeks from t0 to t21. The PLS-DA score plot at t0 and t21 (Figure 3A) showed that animals were divided by sex at t0 (LV1, 18.33%) and by diet at t21 (LV1, 22.03%). Twenty-one weeks of HFD generated a defined metabolic signature in urine (Figure 3B). Hippurate, the most relevant metabolite, decreased significantly from week 3 in both sexes and remained low until week 21 (Figure 3C). N-Phenylacetylglycine (Figure 3D) and indole-3-acetate (Figure 3E) decreased significantly in HFD females from week 3 but not in HFD males. These metabolites are related to host–microbiota co-metabolism [10]. Citrate (Figure 3F) and 2-oxoglutarate (Figure 3G) also decreased from week 3 in HFD animals, though not always significantly. Glycine increased in HFD males from week 3 but showed high variability over time (Figure 3H).

### 2.5. HFD Induces Extensive and Sex-Dimorphic Changes in Faecal Metabolome as Early as Week 3

The faecal metabolome was analysed in male and female Wistar rats every three weeks from t0 to t21. The PLS-DA score plot at t0 and t21 (Figure 4A) clearly showed a differentiated distribution of the groups at t21. Twenty-one weeks of HFD were sufficient to generate a strong metabolic signature in faeces among CTL and HFD groups (Figure 4B). Faecal samples exhibited the most differences compared to other sample types. First, short-chain fatty acids (SCFAs) such as acetate (Figure 4C), butyrate (Figure 4D), and propionate (Figure 4E), were statistically decreased from week 3. Second, branched-chain amino acids (BCAAs) like valine (Figure 4F), leucine (Figure 4G), and isoleucine (Figure 4H) were statistically increased from week 3, with greater differences observed in HFD males compared to HFD females. Third, cadaverine showed a statistical increase only in HFD males at t3, remaining stable until the endpoint, while in females, the significant increase was observed from t6 to t15 (Figure 4I). Additionally, homovanillate (Figure 4J) and tau-methylhistidine (Figure 4K) statistically decreased from t3 to t21 in HFD animals. Choline (Figure 4L) and thymine (Figure 4M) significantly increased from t3, remaining stable until t21 in HFD animals. SCFAS, BCAAS, cadaverine, homovanillate, and choline are directly linked to host–microbiota co-metabolism [9,10].

### 2.6. Decreased Faecal Microbiota Diversity and Lactobacillus Content After 3 Weeks of HFD Precedes Host Metabolomic Changes

Given the large number of metabolites related to host–microbiota co-metabolism, an exclusive study of the faecal microbiota was performed. By week 3, there was already a measurable change in the faecal microbiota, differentiating CTL groups from HFD animals (Figure 5A,B). The most striking alteration was observed in *Lactobacillus sp*., which almost disappeared from week 3 in HFD animals and remained decreased until the experiment ended (Figure 5B,C). An increase in *Lachnospiraceae* was observed from week 3 in HFD animals, mainly maintained over time in HFD males (Appendix A). No statistically significant differences were observed in *Bacteroidaceae* and *Prevotellaceae* (Appendix A). A significant increase in *Porphyromonadacea* and *Tannerellacea* was observed in HFD males from week 9 and in HFD females from week 12 (Appendix A). Moreover, a significant increase over time was observed in *Anaeroplasma* sp. (Appendix A) and *Clostridiaceae, Eubacteriaceae, and other families* (Appendix A), mainly in HFD males. The Firmicutes to Bacteroidetes (F/B) ratio significantly increased in HFD animals at weeks 18 and 21 (Figure 5D). Cluster analysis of faecal microbiota and metabolomics results among groups and over time (Figure 6A) revealed changes in faecal microbiota at t3, also observed in co-metabolites in faecal and urine samples, suggesting the important role of the microbiota in changes preceding the early subclinical stages of MASLD. Moreover, this pattern was confirmed by a Spearman correlation heatmap, where we observed strong positive (red) and negative (blue) correlations between metabolites in serum, faeces, and urine with changes in the gut microbiota starting from week 3 (Figure 6B). By week 21, the correlation pattern was maintained and became more pronounced. Additionally, variables such as the percentage of fat quantified by ORO staining in the liver and BA levels in plasma were included at week 21. It was observed that a lower presence of *Lactobacillus* in the microbiota positively correlated with a higher percentage of fat in the liver (Appendix A).

## 3. Discussion

Early detection of MASLD is crucial, as the condition is reversible at early stages with lifestyle interventions [2]. Although different studies show associations between established MASLD and gut microbiota alterations [5], it is unclear if it plays any role in early stages. In this study, we induced early subclinical MASLD in male and female Wistar rats. After 21 weeks of HFD, our model demonstrated sex-specific and systemic metabolic alterations. In HFD animals, histological detection of early-stage hepatic steatosis supported the existing subclinical stage of MASLD. In addition, HR-MAS metabolomics revealed shifts in hepatic metabolism beyond fatty acid pathways and highlighted that HFD not only promoted fat accumulation in the liver but also altered the composition of lipid species toward more harmful profiles. Additionally, a general reduction in plasma BA content was observed in HFD-treated animals. This reduction in circulating BA levels has been linked to disruptions in lipid metabolism and gut microbiota associated with metabolic disorders [11]. Then, to study the progression of this diet-induced fat accumulation, we closely and extensively monitored changes in faecal microbiota and blood, urine, and faecal metabolomes every three weeks. For the first time to our knowledge, we demonstrate that microbiota and microbial co-metabolism alterations, which happen as early as after only 3 weeks of HFD, precede host metabolism changes since most of the significant metabolites are related to bacterial metabolism. Our results also provide many metabolomic differences at different time points between rats fed with a control diet and an HFD in blood, urine, and faecal extracts. All these HFD-induced changes were sex dimorphic with more overall detrimental effects for males than females. These findings may help to identify individuals with early subclinical fatty liver.

Liver intact tissue metabolomics reveals substantial changes not only in the amount but also in the profile of liver lipids in Wistar rats after 21 weeks of HFD. Overall, the amount of polyunsaturated fats decreased whereas the saturated and monounsaturated fats, more detrimental, increased. Saturated fats can induce lipotoxicity, causing cellular damage and inflammation in liver cells [12], while a decrease in PUFAs reduces their protective anti-inflammatory effects [3]. MUFAs, major contributors to the lcUFA group, have been reported to increase in MASLD [3]. The decrease in PUFAs, also reflected in a decrease in tUFAs, is expected after HFD and has been associated with MASLD severity [3,13]. Additionally, reduced PUFAs can disrupt insulin sensitivity, further compromising liver health and function [14]. We also observed an increase in lipid species rich in carbonyl moieties possibly reflecting increased lipid oxidation and peroxidation [15]. All these imbalances can lead to increased oxidative stress and impaired lipid metabolism, contributing to hepatic steatosis and metabolic dysfunction.

Blood metabolomics is ideally suited for monitoring host metabolism as it provides a comprehensive snapshot of systemic metabolic processes, reflecting the body’s internal biochemical state. Faecal metabolomics, on the other hand, excels at capturing the metabolic activities of gut microbiota [10], offering detailed insights into the complex interactions between the microbiome, diet, and host. At an intermediate, urine metabolomics reflects both host and microbial metabolism, as urine contains metabolites excreted from both sources, providing a broader perspective on metabolic health [10,16]. Our results reveal moderate changes in blood metabolome after 21 weeks of HFD, suggesting minor alterations in host metabolism. The main changes observed were reflected in metabolites associated with bacterial metabolism (TMAO, choline, and N-acetyl compounds) and fat metabolism (choline, PUFAS, and tUFAs). At the other side, faecal and urine metabolome were significantly altered after only 3 weeks of HFD. In faeces, it is straightforward to observe changes in bacterial metabolism since the microbiota is the first to be exposed to dietary changes and will adapt its metabolism to the new nutrients. Changes in SCFAs, choline, cadaverine, homovanillate, among others, have been associated with alterations in bacterial diversity, bacterial metabolism, and MASLD [10]. In urine, the significant metabolites (hippurate, indoles, N-phenylacetylglycine) were also related to changes in gut microbiota. The only difference is that these metabolites reached the liver, were metabolised, and then excreted in urine [10]. All these metabolic hints point towards early changes in faecal microbiota ecosystems. Moreover, our microbiota characterisation confirms that faecal microbiota changes after only 3 weeks of HFD in both male and female Wistar rats. There is a strong decrease in the *Lactobacillus* genus observed from week 3, which does not recover over time and can be associated with and correlated to metabolic detrimental effects. Our rats also exhibited an increased Firmicutes to Bacteroidetes (F/B) ratio over time with HFD, consistent with previous studies in MASLD [5]. This result suggests a shift towards a microbiota composition that could be favouring a metabolism with energy extraction and fat deposition in the liver. Although both male and female Wistar rats showed changes in bacterial diversity in faecal microbiota after HFD, the effect is sex-dimorphic, with females exhibiting a more dynamic faecal microbiota than males. Our findings suggest that HFD induce detrimental alterations in gut microbiota, which affect microbial metabolism after a short time and much earlier than those in the host metabolism.

Our study provides many potential biomarker candidates for subclinical MASLD and for individuals at risk. Although blood metabolome shows smaller HFD impact, the changes identified are still reflecting a subclinical fat accumulation in the liver and a change related to microbial metabolism. At week 21, we observed some blood metabolome changes closely related to those identified in liver tissue and involving fatty acid species (PUFAs, tUFAs) and some metabolites related to host–microbiota co-metabolism (TMAO and choline compounds). Interestingly, N-acetyl groups, which whose supplementation exhibits anti-inflammatory effects and may reduce the onset of steatosis [17,18], are also decreased in blood after 21 weeks of HFD. On the other hand, we observed urine metabolome alterations related to amino acid metabolism, tricarboxylic acid (TCA) cycle, and host–microbiota co-metabolism, after only 3 weeks of HFD and generally showing larger effects in female than in male Wistar rats. Hippurate and N-phenylacetylglycine are glycine-conjugated compounds related to host–microbiota co-metabolism, both decreased in MASLD [10,19]. Additionally, high levels of hippurate have been suggested as an overall indicator of microbial diversity and metabolic health [20,21]. Finally, faecal content in SCFAs, BCAAs, and microbiota metabolites is dramatically altered towards detrimental profiles also after only 3 weeks of HFD, with larger effects in male than in female Wistar rats. Higher levels of SCFAs are associated with healthy microbiota, regulate host metabolic processes, and control faecal pH [22,23]. Elevated levels of BCAAs have been associated with MASLD, type 2 diabetes mellitus (T2DM), and insulin resistance (IR) [24]. All these metabolic changes have been associated to metabolic disease or related processes in the past and are also affected in our study in early stages of MASLD progression.

Throughout this study, we have identified numerous metabolites associated with bacterial metabolism that were found to be altered across different types of samples. Understanding these microbial co-metabolism shifts opens new avenues for early intervention and understanding in the MASLD context. Preventive and therapeutic strategies for MASLD could include dietary modifications to increase fibre intake [4] and the use of probiotics to restore beneficial bacteria and prebiotics to support a healthy microbiota balance [5]. Probiotics such as *Lactobacillus* and *Bifidobacterium* have shown promise in restoring gut microbiota balance and improving liver health. The supplementation of *Lactobacillus* in a Western diet mouse model reduced the progression of steatosis in liver disease, being proposed as a beneficial strategy to treat MASLD [25]. Additionally, lifestyle changes such as regular physical activity and reducing the intake of saturated fats can help mitigate the adverse effects of Western diets [3].

We acknowledge several limitations in our study. First, while Wistar rats are valuable for studying MASLD stages, there are limitations when extrapolating these findings to humans. Physiological differences, such as metabolic rate and immune responses, can affect disease progression and treatment outcomes. Additionally, the genetic diversity in humans is much greater than in inbred rat strains, leading to varied susceptibilities and responses. The controlled environment and standardised diet of laboratory rats do not fully replicate the complex dietary and environmental influences experienced by humans. Another limitation of the study is that the analysis of the faecal microbiota was performed using qPCR instead of 16S rRNA sequencing or metagenomics techniques. Both qPCR and sequencing are valuable tools for analysing gut microbiota, but they have different strengths. qPCR is highly specific and sensitive, making it ideal for quantifying known microbial species. It is also faster and more cost-effective, which makes it particularly suitable for longitudinal studies. On the other hand, sequencing provides a comprehensive view of the entire microbial community, allowing for the identification of novel species and offering higher taxonomic resolution. We chose qPCR for its accuracy, quantitative consistency, cost-effectiveness, and translational potential. This technique enables the quantification of specific microbial species by targeting their unique genetic markers. Due to its high sensitivity, qPCR is an effective method for monitoring changes in the composition and abundance of gut microbiota. Based on the literature, we selected a representative set of bacterial families and genera to be quantified using qPCR. The consistent results obtained over time in our study using extensively validated primers in qPCR technique suggest that the qPCR data were sufficient to establish a sequence of events and validate our study. Another limitation is that organs were collected only at the end and not at intermediate points, preventing us from assessing longitudinal changes in hepatic metabolomics. We also acknowledge that our model reflects only the early onset of liver steatosis, which represents an initial stage of MASLD. Nevertheless, we believe this is precisely one of the key strengths of our study, as steatosis is a reversible stage. Finally, mechanistic studies are needed to integrate the progression of changes.

## 4. Materials and Methods

### 4.1. Animals and Housing

Seventeen-week-old male (518.1 ± 32.7 g) and female (262.2 ± 13.4 g) Wistar rats (JanvierLabs, Le Genest-Saint-Isle, France) were housed under a 12-h light-dark cycle, with constant temperature and humidity (22 ± 2 °C and 55%, respectively), and provided with ad libitum access to food and water. After a one-week acclimatisation period, the animals were randomly assigned to four experimental groups: chow diet (CTL) males (*n* = 8), CTL females (*n* = 8), high-fat diet (HFD) males (*n* = 10), and HFD females (*n* = 10). The animals were fed either a CTL diet (2014S, ENVIGO, West Lafayette, Indiana, USA) or a 45% HFD (TD.08811, Ssniff, Soest, Germany) for 21 weeks. Serum, urine, and faeces samples were collected every three weeks, defining the experimental time points (in weeks) t0, t3, t6, t9, t12, t15, t18, and t21. Intraperitoneal glucose tolerance tests (IP GTT) were performed at weeks 12 and 21. After 21 weeks, the animals were sacrificed by inhalation of 5% isoflurane under fasting conditions. Post-mortem procedures are detailed in the Appendix A and Methods. All animal procedures were supervised and approved by the Ethics Committee in Experimental Research of the University of Valencia (2019/VSC/PEA/0129-A1538561308126/type2).

### 4.2. Histopathological Assessment

Liver histology was assessed using various staining techniques. Haematoxylin-eosin (H&E) staining was used for general tissue visualisation, and Masson’s trichrome staining was used for collagen fibre detection. These stains were performed on paraffin-embedded liver sections, cut into 5 µm slices using a microtome (Leica Microsystems, Wetzlar, Germany). Oil Red O (ORO) staining, used for lipid droplet visualisation, was performed on non-paraffin-embedded frozen liver sections, cut into 4–7 µm slices with a cryostat (CM1900, Leica Microsystems, Wetzlar, Germany). All staining reagents were obtained from Sigma-Aldrich, Darmstadt, Germany. Photographs of all stains were taken with a digital microimaging device (DMD108, Leica Microsystems, Wetzlar, Germany). ORO staining was quantified using ImageJ2 software (accessed on 1 March 2024 https://imagej.net/software/imagej/). Five random photographs per liver section were analysed, applying the same macro to quantify red pixels with the following settings: “Color Threshold tool”: 0–255 Hue, 120–255 Saturation, 200–255 Brightness.

### 4.3. Biochemical Analysis of Plasma

Biochemical analysis of plasma was performed at the endpoint (week 21). During animal sacrifice, 500 μL of whole blood was collected and preserved in EDTA (0.4 M, pH 8) (E6511-100G, Sigma, San Luis, MO, USA). Albumin (ALB), alkaline phosphatase (ALP), alanine aminotransferase (ALT), aspartate aminotransferase (AST), bile acid (BA), total cholesterol (CHOL), and urea were measured in 200 µL of plasma using a Skyla VB1 biochemistry analyser (Skyla Corporation, Taiwan) with Liver Plus-12 rotors (900-180, Skyla Corporation, Taiwan). White blood cells (WBCs), neutrophils, red blood cells (RBCs), and platelets (PLT) were measured in 15 µL of plasma using an Element HT5 analyser (Heska Corporation, Loveland, CO, USA). The AST/ALT ratio was calculated.

### 4.4. Metabolomics Using Proton Nuclear Magnetic Resonance (^1^H-NMR)

A Bruker AVANCE III NMR spectrometer (Bruker BioSpin GmbH, Rheinstetten, Germany) operating at 600.13 MHz proton (^1^H)–NMR frequency was used to measure the samples. For serum, urine, and faecal extracts, 20 µL were introduced into 1mm NMR capillary tubes (Z107504, Bruker, Karlsruhe, Germany) and measured using a 1 mm triple resonance (TXI) probe of ^1^H-NMR. Livers stored at −80 °C were fractionated with liquid nitrogen. Fragments (50–60 mg) were collected, and placed into zirconia rotors (HZ07213, Bruker, Karlsruhe, Germany). Then, 40 µL of sterile deuterium oxide (D2O) (1.13366, Sigma, San Luis, MO, USA) were added, and the rotors were sealed. The rotors were manually introduced into the ^1^H-NMR equipment. High-Resolution Magic Angle Spinning (HR-MAS) ^1^H-NMR probe was used for measurement. Spectra were processed using MestReNova software (MestReNova v14.1.1, Mestrelab Research S.L, Santiago de Compostela, Spain) for phase, baseline, and reference correction. Chenomx software (Chenomx NMR Mixture Analysis v8.1, Edmonton, AB, Canada) was used to assign metabolites to each peak. Databases such as PubMed, Human Metabolome Database (HMDB), and Kyoto Encyclopaedia of Genes and Genomes (KEGG) were used to confirm metabolites. The listed metabolites appear in Appendix A. MATLAB software (MATLAB R2014a, MathWorks, Natick, MA, USA) was used to obtain the Partial Least-Squares Discriminant Analysis (PLS-DA) model and Variable Importance in the Projection (VIP) scores. Extended sample preparation, ^1^H-NMR methodology, and spectral analysis are detailed in the Appendix A and Methods.

### 4.5. DNA Extraction from Stool Samples

Bacterial genomic DNA from faecal samples frozen at −80 °C was extracted using the QIAamp Fast DNA Stool Mini Kit (51604, QIAGEN N.V., Hilden, Germany) with certain modifications. Briefly, faeces (180–220 mg) underwent homogenisation in 1mL of InhibitEX buffer and were left at 60 °C overnight. The following day, samples were centrifuged at 14.000 rpm for 12 min at 4 °C. Supernatant was collected in a new Eppendorf tube and 25 µL of Proteinase K plus 600 µL of AL buffer were added. The mixture was incubated for 15 min at 70 °C under shaking conditions. Following that, 600 µL of cold 96% ethanol was introduced, and samples underwent centrifugation at 14.000 rpm for 2 min at 4 °C. Afterward, 500 µL of AW1 buffer was added and centrifuged again at 14.000 rpm for 2 min at 4 °C. Subsequently, 500 µL of AW2 buffer was added and two successive centrifugations were executed at 14.000 rpm for 2 min at 4 °C. DNA elution from columns was carried out in three consecutive steps in which 60 µL of ATE buffer was added, columns were left to incubate at room temperature (40 min, 15 min, 15 min respectively) and centrifuged at 8.000 rpm for 3 min at 4 °C. Altogether, 180 µL containing genomic DNA were acquired.

### 4.6. Microbial Analysis by Quantitative PCR

Quantitative PCR (qPCR) was done in 384-well microplates (4309849, Applied Biosystems, Carlsbad, CA, USA) sealed with an adhesive film (4311971, Applied Biosystems, USA). qPCR was carried out using a Power track SYBR Green Master mix (A46110, Applied Biosystems, USA) and QuantStudioTM 5 Real-Time PCR system (Applied Biosystems, USA). Reactions mixtures (total of 10 µL) included 5 µL of Power track SYBR Green Master mix, 0.4 µL of a primer pair final concentration of 0.2 mM, 3.6 µL of molecular water, and 1 µL of DNA (from frozen aliquots of 20 ng/µL), in accordance with the recommendations of Marques, C et al. [26]. Negative controls had all the components except 1 µL of DNA, which was substituted for 1 µL of molecular water (BP2819-1, Fisher Scientific, Waltham, MA, USA). The primer BLAST software was used to design group-specific primers for microbial detection targeting the 16S rRNA gene (Appendix A), and their target species (Appendix A). Additionally, some primers were taken from the following already published articles: [26,27,28]. To ascertain the specificity of the amplicon, a melting curve analysis occurred during the final stage of the qPCR, and the amplification products of each primer pair were analysed in an electrophoresis 2–3% agarose gel (8016, Conda Pronadisa, Madrid, Spain). PCR templates and data underwent processing and analysis using QuantStudio™ Design & Analysis software v1.5.3 (Applied Biosystems, Carlsbad, CA USA). Each sample was run per triplicate. Ct, ∆Ct, Δ∆Ct, 2^−Δ∆Ct^ values were used to calculate the proportion of bacterial taxa in faeces. All the data were standardised to CTL male values, and the positive control measured by Universal 16S rRNA 520F-799R primer (Appendix A). Data were presented as the relative abundance of the analysed bacteria. Firmicutes/Bacteroidetes (F/B) ratio was calculated using Lachnospiraceae family and Bacteroidaceae and Prevotella families qPCR data, because these groups are considered references of Firmicutes and Bacteroidetes respectively [5]. DNA purity (260/280 and 260/230 ratio) and concentration (ng/µL) were measured with a NanoDrop 2000 spectrophotometer (ND-2000, ThermoFisher, Waltham, MA USA). Aliquots of genomic DNA at 20 ng/µL were generated for quantitative PCR following the recommendation of Marques, C et al. [26], and stored at −80 °C until use.

### 4.7. Statistical Analysis and Biological Interpretation

Statistical analysis was performed using SPSS software (IBM SPSS Statistics 28.0, New York, NY, USA). Normality was evaluated using the Kolmogorov–Smirnov test. For normally distributed variables, parametric tests (Student’s *t*-test (pairwise analysis), two-way ANOVA, or factorial ANOVA for related samples) were applied. For non-normally distributed variables, non-parametric tests (Friedman, Wilcoxon, and Mann–Whitney U) were used. Only microbiota data related to the Lactobacillus genus did not follow a normal distribution, and non-parametric tests were applied. Intra-subject variability was assessed by linear regression, and housing effects by cage were evaluated using ANOVA as an additional variable. One CTL male and one HFD female were removed from the study as outliers. The Spearman correlation analysis was run using MetaboAnalyst 6.0 (accessed on 1 March 2024, https://www.metaboanalyst.ca/) to analyse the relationship between the different metabolites found in faeces, serum, and urine and the gut microbiota data. Statistical significance was set at *p* < 0.05. Further details on statistical analysis are provided in the Appendix A and Methods.

## 5. Conclusions

In conclusion, our study highlights that alterations in gut microbiota co-metabolism precede host metabolic changes during the early onset of liver steatosis, a subclinical and reversible stage of MASLD. A three-week HFD was sufficient to induce shifts in gut microbiota, evidenced by changes in bacterial metabolism and specific bacterial species. Metabolites linked to bacterial metabolism were identified in serum, faeces, urine, and liver samples, suggesting that these microbial metabolic changes may precede and contribute to host metabolic alterations and the progression of MASLD. Furthermore, histological analysis and liver metabolomics revealed sex-specific differences in hepatic fat accumulation and lipid composition induced by HFD, despite the absence of detectable changes in blood liver function markers. Notably, a significant reduction in circulating BAs was observed in HFD-treated animals, a feature associated with gut microbiota alterations and lipid metabolism disruptions characteristic of metabolic disorders. The identification of sex-specific metabolic signatures in serum, urine, and faeces provides potential proxy markers for early detection of the early-onset of liver steatosis disease and at-risk individuals. Our results may also open new perspectives for developing new preventive strategies.

## Figures and Tables

**Figure 1 ijms-26-01288-f001:**
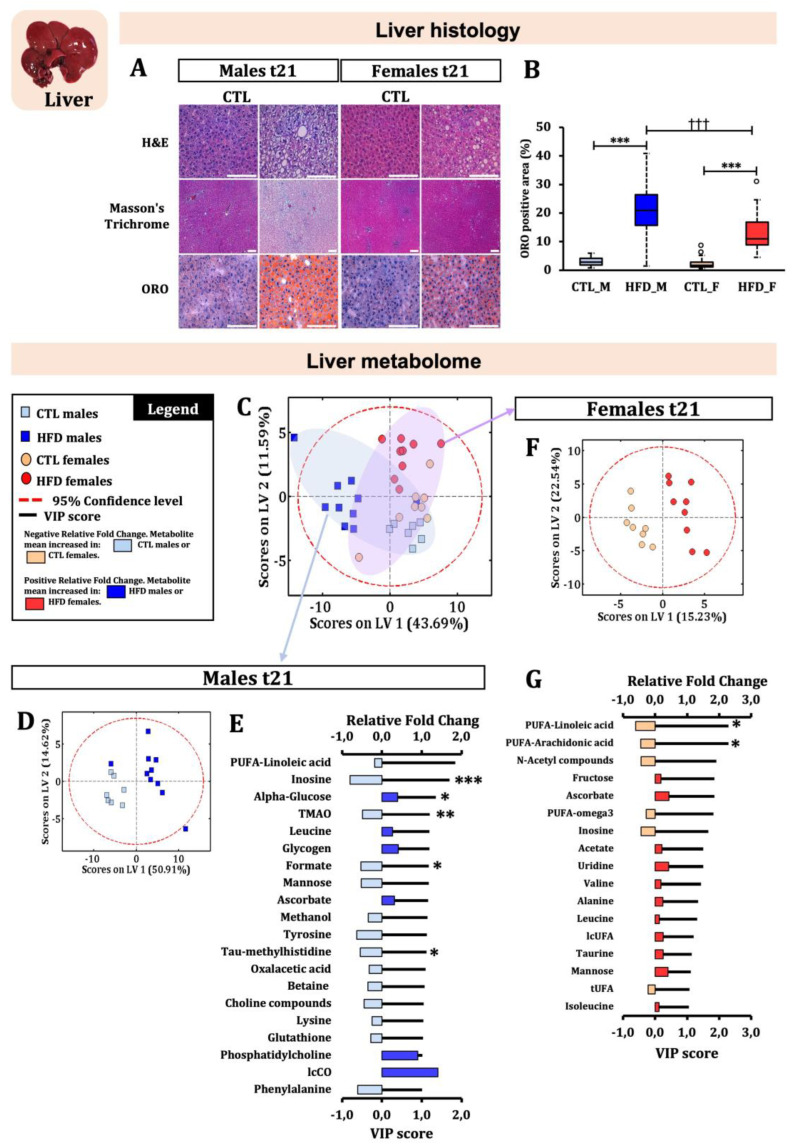
**Liver histopathological and metabolomic assessment in male and female Wistar rats after 21 weeks of HFD.** (**A**) Representative liver sections in males and females at t21. Haematoxylin and Eosin (H&E) and Oil Red O (ORO) images were at 40× magnification, while Masson’s Trichrome stain was at 10× magnification (the scale appears as a white line at the bottom of the images). (**B**) ORO-positive area (%) quantification by ImageJ at t21. Statistical significance was set at *** *p* < 0.001 (CTL vs. HFD groups), and ^†††^
*p* < 0.001 (males vs. females) (two-way ANOVA and Bonferroni post hoc test). PLS-DA score plot showing (**C**) all the groups, (**D**) males, and (**F**) females. (**E**) VIP score and relative fold change plot shows metabolites which present a VIP score > 1 obtained from the PLS-DA score plot from (**D**), and the increase or decrease in the mean of each metabolite (relative fold change) from HFD males compared to CTL males. (**G**) Idem as (**E**) but in females. (**E**,**G**) Statistically significant differences (Student’s *t* test with adjusted *p*-value) were set at * *p* < 0.00081, ** *p* < 0.00016, and *** *p* < 0.000016. Relative fold change was calculated for each metabolite using the following ratio: [(HFD animal mean − CTL animal mean)/CTL animal mean]. VIP score values were represented by black lines. Relative fold change values were represented by bars. Abbreviations: CTL_F, CTL females; CTL_M, CTL males; HFD_F, HFD females; HFD_M, HFD males; lcCO, long-chain carbonyl groups; lcUFA, long-chain unsaturated fatty acid; PUFA, polyunsaturated fatty acid; TMAO, trimethylamine N-oxide; tUFA, total unsaturated fatty acid.

**Figure 2 ijms-26-01288-f002:**
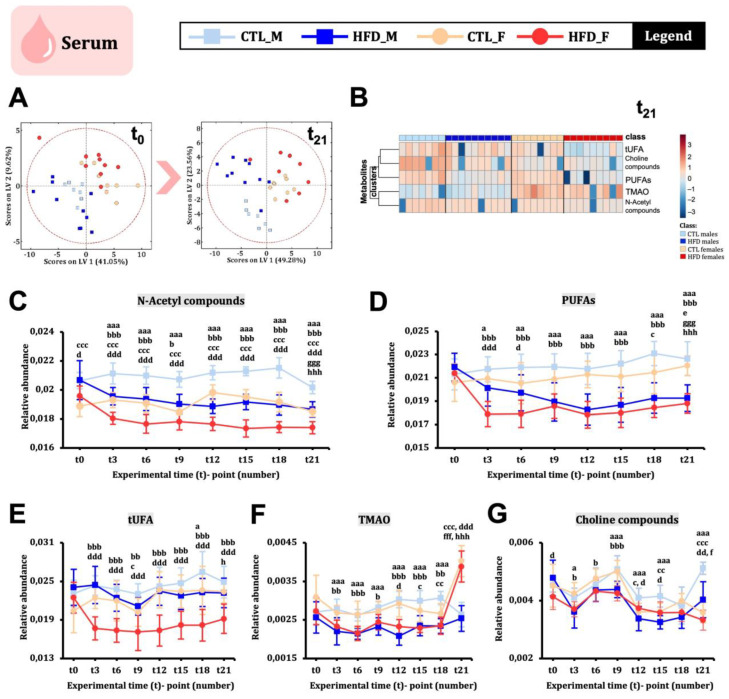
**Serum metabolites changes across 21 weeks of HFD in male and female Wistar rats.** (**A**) PLS-DA score plots at t0 and t21. (**B**) Heatmap built using MetaboAnalyst and showing the metabolites presented from (**C**–**G**) with the data at week 21. In the heatmap, the colour showed the metabolite concentration in each animal condition from highest (red) to lowest (blue). (**C**–**G**) Serum metabolite evolution over time. Statistical significance was set at *p* < 0.05, *p* < 0.01, or *p* < 0.001 (factorial ANOVA and Bonferroni post hoc test). For between-group comparisons, the following letter code was used: a, CTL male vs. HFD male; b, CTL female vs. HFD female; c, CTL male vs. CTL female; d, HFD male vs. CTL female; e, CTL male t0 vs. CTL male t21; f, HFD male t0 vs. HFD male t21; g, CTL female t0 vs. CTL female t21; h, HFD female t0 vs. HFD female t21. Abbreviations: PUFA, polyunsaturated fatty acid; TMAO, trimethylamine N-oxide; tUFA, total unsaturated fatty acid.

**Figure 3 ijms-26-01288-f003:**
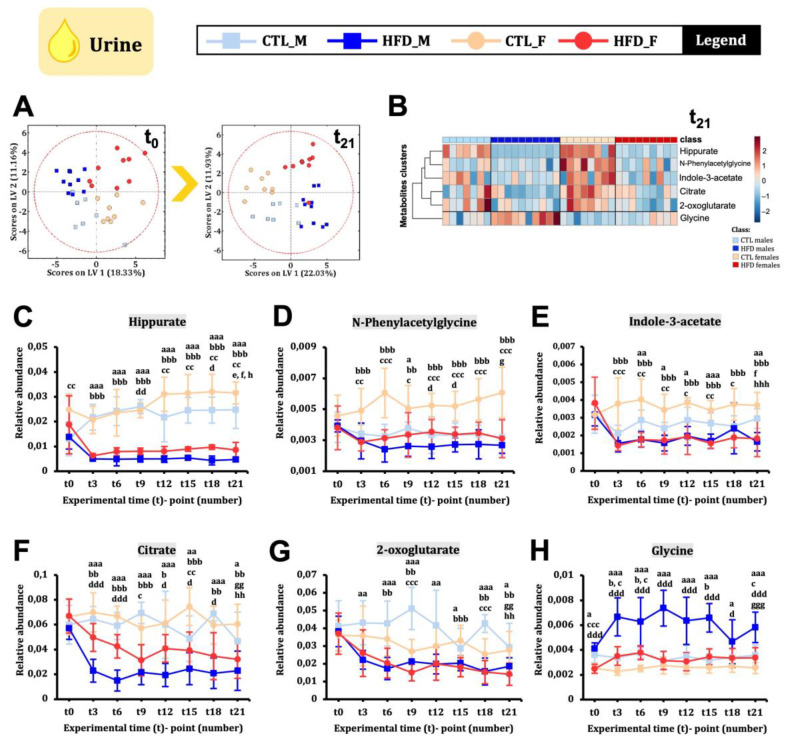
**Urine metabolites changes across 21 weeks of HFD in male and female Wistar rats.** (**A**) PLS-DA score plots at t0 and t21. (**B**) Heatmap showing the metabolites presented from (**C**–**H**) with the data at week 21. In the heatmap, the colour showed the metabolite concentration in each animal condition from highest (red) to lowest (blue). (**C**–**H**) Urine metabolites evolution over time. Statistical significance was set at *p* < 0.05, *p* < 0.01, or *p* < 0.001 (factorial ANOVA and Bonferroni post hoc test). For between-group comparisons, the following letter code was used: a, CTL male vs. HFD male; b, CTL female vs. HFD females; c, CTL male vs. CTL female; d, HFD male vs. CTL female; e, CTL male t0 vs. CTL male t21; f, HFD male t0 vs. HFD male t21; g, CTL female t0 vs. CTL female t21; h, HFD female t0 vs. HFD female t21.

**Figure 4 ijms-26-01288-f004:**
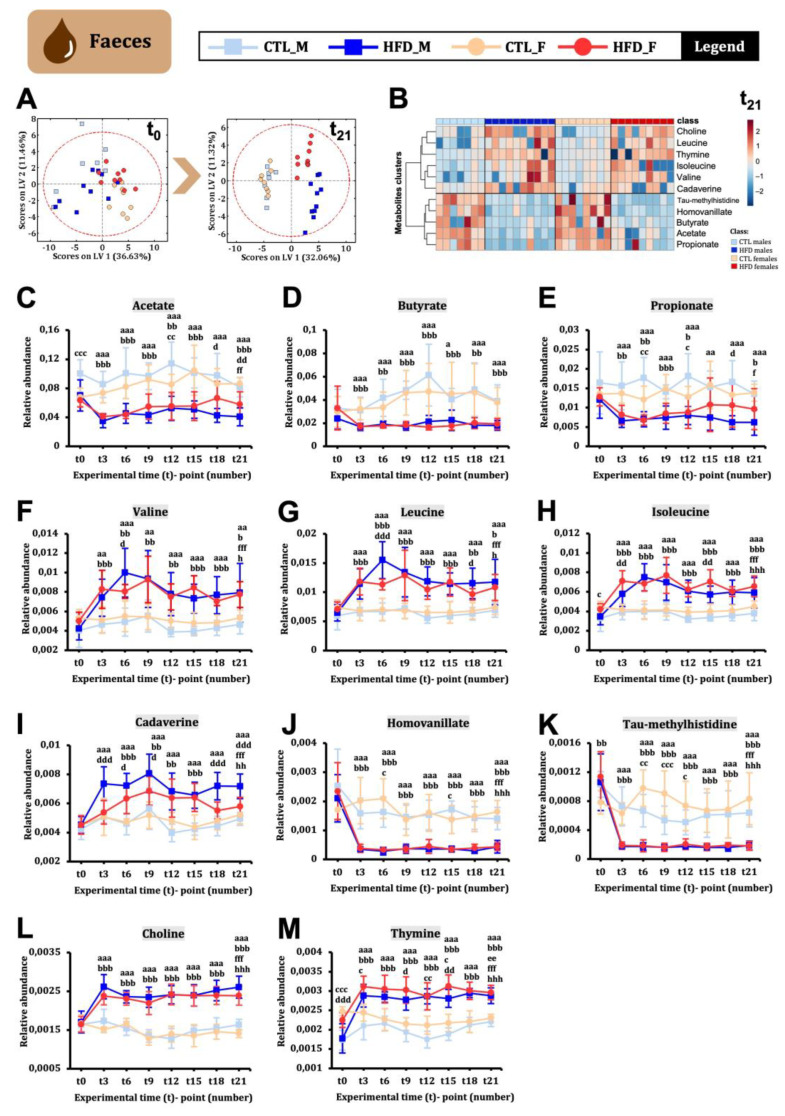
**Faecal metabolites changes across 21 weeks of HFD in male and female Wistar rats.** (**A**) PLS-DA score plots at t0 and t21. (**B**) Heatmap showing the metabolites presented from (**C**–**M**) with the data at week 21. In the heatmap, the colour showed the metabolite concentration in each animal condition from highest (red) to lowest (blue). (**C**–**M**) Faecal metabolites evolution over time. Statistical significance was set at *p* < 0.05, *p* < 0.01, or *p* < 0.001 (factorial ANOVA and Bonferroni post hoc test). For between-group comparisons, the following letter code was used: a, CTL male vs. HFD male; b, CTL female vs. HFD females; c, CTL male vs. CTL female; d, HFD male vs. CTL female; e, CTL male t0 vs. CTL male t21; f, HFD male t0 vs. HFD male t21; g, CTL female t0 vs. CTL female t21; h, HFD female t0 vs. HFD female t21.

**Figure 5 ijms-26-01288-f005:**
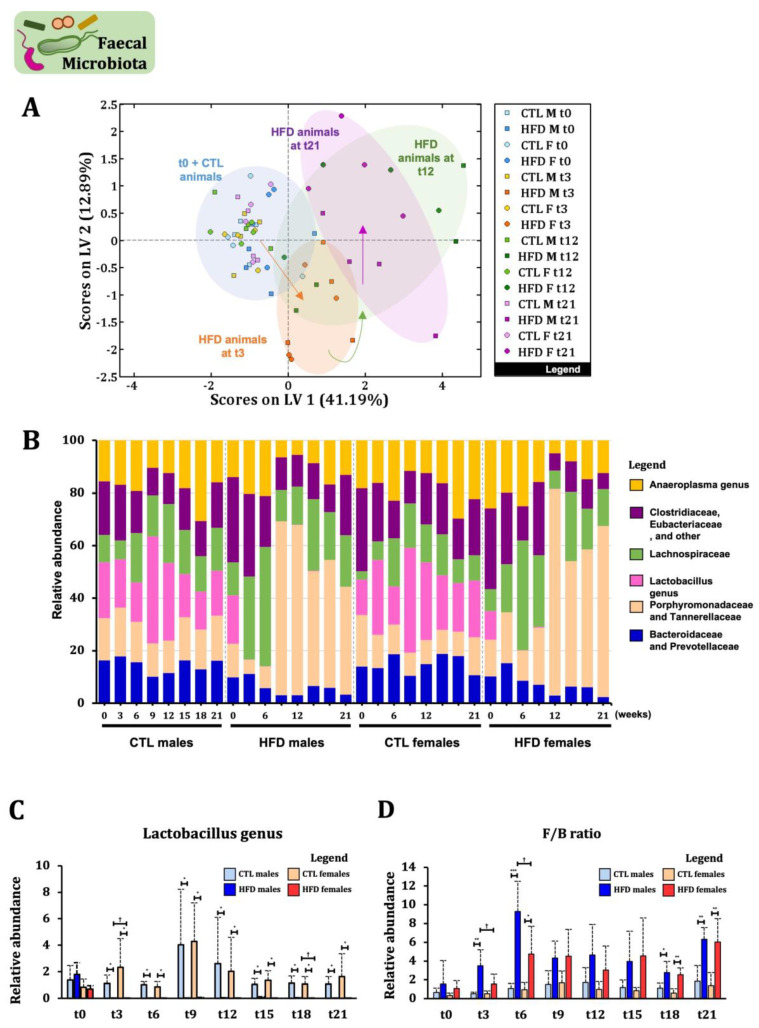
**Faecal microbiota changes across 21 weeks of HFD in male and female Wistar rats.** (**A**) PLS-DA score plot among the faecal microbial data at week 0, 3, 12, and 21 (*n* = 4 animals per group). (**B**) Relative abundance of specific faecal bacterial groups over 21 weeks. Each bar represents the average relative abundance at specific group level in the faecal microbiota among groups. Relative abundance of (**C**) Lactobacillus genus and (**D**) Firmicutes/Bacteroidetes (F/B) ratio. Statistical significance was set at * *p* < 0.05, ** *p* < 0.01, *** *p* < 0.001 (CTL vs. HFD group) and ^†^
*p* < 0.05 (males vs. females). Friedman test for repeated measures and Mann–Whitney U test were applied in (**C**) while factorial ANOVA and Bonferroni post hoc test in (D). *n* = 4 animals per group.

**Figure 6 ijms-26-01288-f006:**
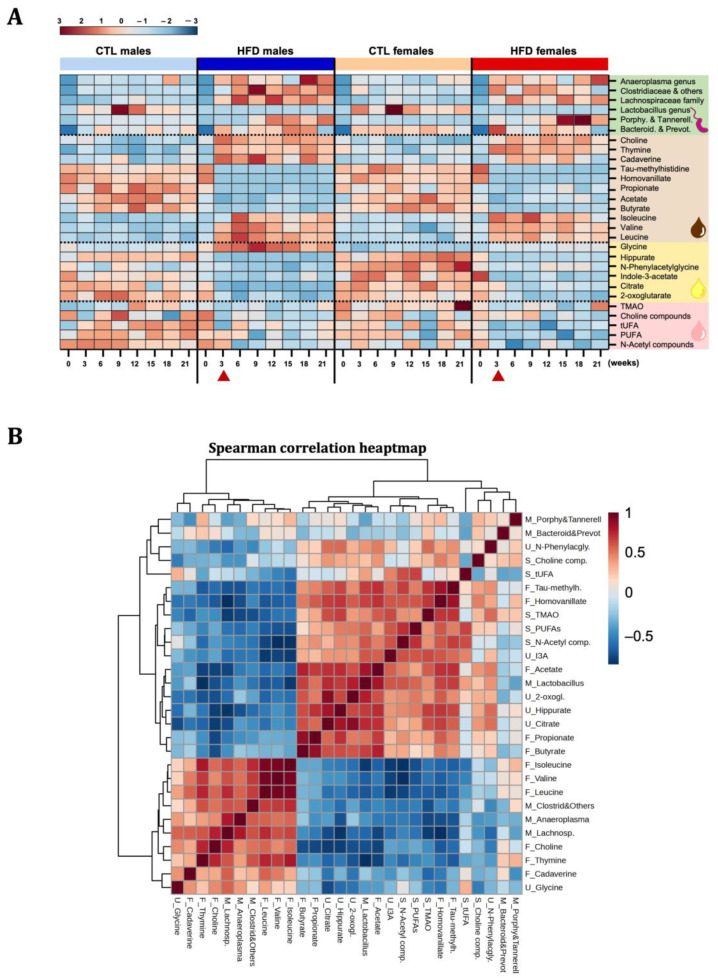
**Cluster analysis.** (**A**) Heatmap showing clustering of microbiota results (from Figure 5) and relevant metabolites seen in serum (from Figure 2), urine (from Figure 3), and faeces (from Figure 4), during CTL or 45% HFD exposure, in males and females over time. Each square represents the average for this metabolite/microbe at this time point. Red triangles highlight changes at t3 in the microbiota and metabolomic pattern in HFD males and HFD females. (**B**) Spearman correlations of the metabolic factors with relative abundance of microbial species at week 3. The factors were abbreviated as M_ for microbiota data, S_ for serum, F_ for faeces, and U_ for urine. Abbreviations: 2-oxogl, 2-oxoglutarate; I3A, indole-3-acetate; N-acetyl comp, N-acetyl compounds; N-phenylacgly, N-phenylacetylglycine; PUFA, polyunsaturated fatty acid; Tau-methylh, tau-methylhistidine; TMAO, trimethylamine N-oxide; tUFA, total unsaturated fatty acid.

**Table 1 ijms-26-01288-t001:** Anthropometrical, haematologic, and biochemical parameters of male and female Wistar rats at endpoint.

Parameters	Males		Females	
	CTL (*n* = 7)	HFD (*n* = 10)	CTL (*n* = 7)	HFD (*n* = 9)
** *Anthropometrical determinations* **				
Energy intake (Kcal/day)	76.8 ± 4.5	109.2 ± 16.3 ***	47.0 ± 3.1 ^†††^	59.5 ± 9.6 ***^†††^
BW (g)	675.3 ± 42.4	731.4 ± 70.7 *	322.6 ± 17.4 ^†††^	326.6 ± 16.1 ^†††^
BMI (g/cm^2^)	0.93 ± 0.13	1.02 ± 0.09 *	0.55 ± 0.02 ^†††^	0.54 ± 0.04 ^†††^
LW (g)	15.6 ± 0.5	17.9 ± 1.7 ***	7.8 ± 1.0 ^†††^	7.8 ± 0.9 ^†††^
LW/BW ratio (%)	2.3 ± 0.2	2.5 ± 0.2	2.4 ± 0.3	2.4 ± 0.2
** *Blood cell parameters* **				
WBC (×10^9^/L)	10.3 ± 2.4	9.3 ± 2.0	7.2 ± 2.1 ^††^	6.1 ± 1.5 ^††^
Neutrophils (×10^9^/L)	2.3 ± 0.8	1.9 ± 0.3 *	1.2 ± 0.2 ^†††^	1.0 ± 0.3 ^†††^
RBC (×10^9^/L)	8.3 ± 0.4	8.1 ± 0.4	8.1 ± 0.4	7.8 ± 0.4
PLT (×10^9^/L)	875.3 ± 100.6	893.3 ± 103.1	956.9 ± 117.8	903.8 ± 117.0
** *Biochemical parameters related to liver function* **				
ALB (g/dL)	4.0 ± 0.1	4.0 ± 0.1	4.5 ± 0.4 ^†^	4.6 ± 0.3 ^†††^
ALP (U/L)	53.4 ± 8.2	67.8 ± 14.8 **	53.3 ± 2.3	53.6 ± 4.9 ^††^
ALT (U/L)	56.1 ± 7.2	51.5 ± 9.2	46.3 ± 6.1 ^†^	46.6 ± 7.9
AST (U/L)	92.3 ± 15.7	89.1 ± 12.0	78.6 ± 6.2	85.7 ± 15.4
AST/ALT ratio	1.6 ± 0.2	1.7 ± 0.1	1.7 ± 0.2	1.9 ± 0.2
BA (μmol/L)	25.6 ± 12.6	11.4 ± 5.6 **	28.4 ± 10.6	19.5 ± 5.9 *
CHOL (mg/dL)	120.4 ± 19.9	115.6 ± 29.4	78.9 ± 11.5 ^†††^	76.2 ± 23.8 ^†††^
Urea (mg/dL)	34.9 ± 2.4	33.5 ± 5.4	26.3 ± 2.6 ^†††^	25.6 ± 1.3 ^†††^

* ALB, albumin; ALP, alkaline phosphatase; ALT, alanine aminotransferase; AST, aspartate aminotransferase; BA, bile acid; BMI, body mass index; BW, body weight; CHOL, total cholesterol; LW, liver weight; PLT, platelets; RBC, red blood cell; WBC, white blood cell. Statistically significant differences were set at * *p* < 0.05, ** *p* < 0.01, and *** *p* < 0.001 (CTL vs. HFD groups), and at ^†^
*p* < 0.05, ^††^
*p* < 0.01, ^†††^
*p* < 0.001 (males vs. females) (two-way ANOVA and Bonferroni post hoc test).

## Data Availability

The data presented in this study are available on request from the corresponding authors.

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
