# Peer review of "Microbiota Co-Metabolism Alterations Precede Changes in the Host Metabolism in the Early Stages of Diet-Induced MASLD in Wistar Rats"

_ijms, 2025, doi:10.3390/ijms26031288_

Round 1
Reviewer 1 Report
Comments and Suggestions for Authors
This research was to investigate metabolic changes during the early MASLD progression, for identifying the sequence of events and evaluating the impact of sexual dimorphism and the microbiota on the initial stages of MASLD development. The results revealed that three weeks on a HFD reduced the bacterial diversity in the faecal microbiota of Wistar rats, accompanied by changes in the faecal and urine metabolome. This study is well organized and meaningful for research and clinical purpose.
There are some suggestions for the study:
1. In the title, microbiota alterations precede changes in the host metabolism is not convincible, both of gut microbiota and many host metabolites of blood showed changes from the 3rd week, the evidences were not enough. Or the metabolism should be more specific?
2. In line 73, a significant decrease in BA in HFD-fed animals. We also found this phenomenon in our previous study. Based on the important roles of BAs in lipid digestion and metabolism, I suggest to explain the result of BA decrease for host metabolism and cite the paper “Cai H, Zhang J, Liu C, et al. High-Fat Diet-Induced Decreased Circulating Bile Acids Contribute to Obesity Associated with Gut Microbiota in Mice[J]. Foods,2024,13(5)”
3. In Figure1 A, a measuring scale is recommended to take the place of 40X.
4. Like the author mentioned in the discussion part, 16S rRNA sequencing or metagenomics techniques were not employed instead using qPCR is a limitation of the study. Because the main conclusion in the title is about the gut microbiota alteration, and qPCR is obviously less efficient than the high-throughput sequencing in revealing this question. And the comparison in phylum level of the gut microbiota might be not precise enough.
Author Response
Comment 1: This research was to investigate metabolic changes during the early MASLD progression, for identifying the sequence of events and evaluating the impact of sexual dimorphism and the microbiota on the initial stages of MASLD development. The results revealed that three weeks on a HFD reduced the bacterial diversity in the faecal microbiota of Wistar rats, accompanied by changes in the faecal and urine metabolome. This study is well organized and meaningful for research and clinical purpose.
Answer: We thank the reviewer for the thoughtful reviewing, the positive comments, and the appreciation of the relevance.
Comment 2: There are some suggestions for the study: In the title, microbiota alterations precede changes in the host metabolism is not convincible, both of gut microbiota and many host metabolites of blood showed changes from the 3rd week, the evidences were not enough. Or the metabolism should be more specific?
Answer: We thank the reviewer for this comment, which highlights a critical point of the manuscript. We analyze the metabolites in three matrices. We found that most metabolites showing alterations starting at week 3 were present in faecal samples or in urine and not in blood. Urine and faeces more directly reflect changes in the gut microbiota because they are in constant contact with the digestive system where the microbiota reside. Faeces contain a large amount of microbial DNA and metabolites, providing a direct snapshot of the gut microbiome's composition and activity. Similarly, urine can show metabolites produced by microbiota that have been absorbed and processed by the body, offering insights into microbial-host interactions. Therefore, urine and faeces are strongly associated with bacterial metabolism (as explained primarily in references 9, 10, and 12). The blood better reflects host metabolism with fewer influences from microbiota because it represents the systemic metabolic state of the host, rather than localized processes in the gut. Blood metabolites are more reflective of whole-body physiology and less subject to the immediate effects of diet or microbial activity in the gut. Additionally, blood metabolites offer a more stable and consistent measure of metabolic changes, providing a clearer picture of the host's overall metabolic health. If we combine this metabolic analysis with the microbiota data obtained by qPCR, which also showed important changes at week 3, we can assess that microbiota changes precede those observed in blood, which in turn reflect host metabolism. We have added some sentences in the discussion to clarify this point.
Comment 3: In line 73, a significant decrease in BA in HFD-fed animals. We also found this phenomenon in our previous study. Based on the important roles of BAs in lipid digestion and metabolism, I suggest to explain the result of BA decrease for host metabolism and cite the paper “Cai H, Zhang J, Liu C, et al. High-Fat Diet-Induced Decreased Circulating Bile Acids Contribute to Obesity Associated with Gut Microbiota in Mice[J]. Foods,2024,13(5)”
Answer: Thank you very much for recommending this article. In addition to being very interesting, it reflects what we observed in our results, as you mentioned. We have added it as a reference, as it fits perfectly with the content of our article. It appears as reference number 11.
Comment 4: In Figure1A, a measuring scale is recommended to take the place of 40X.
Answer: Thank you for pointing out this issue. It has now been updated. We have also updated the figure legend.
Comment 5: Like the author mentioned in the discussion part, 16S rRNA sequencing or metagenomics techniques were not employed instead using qPCR is a limitation of the study. Because the main conclusion in the title is about the gut microbiota alteration, and qPCR is obviously less efficient than the high-throughput sequencing in revealing this question. And the comparison in phylum level of the gut microbiota might be not precise enough.
Answer: We thank the reviewer for this observation. Indeed, we are aware that this is one of the study's limitations. qPCR and sequencing are both valuable for analyzing gut microbiota, but they have different strengths. qPCR is highly specific and sensitive, making it ideal for quantifying known microbial species, and it is faster and more cost-effective, making it ideal for longitudinal studies. Sequencing offers a comprehensive view of the entire microbial community, identifying novel species and providing higher taxonomic resolution. We chose qPCR for accuracy, quantitative and consistent data analysis, cost-effectiveness and translational potential. This technique allows for the quantification of specific microbial species by targeting their unique genetic markers. qPCR is highly sensitive, making it an effective method for monitoring changes in the composition and abundance of gut microbiota. We selected, according to the literature, rather representative set of bacterial families and genus to be quantified by qPCR, as described in tables S5 and S6. Our results were consistent across all time points adding validity to both the method used and the findings. We have deemed it appropriate to expand the explanation of the study's limitations.
Reviewer 2 Report
Comments and Suggestions for Authors
The authors make the claim that their model reflects early stages of MASH or MASLD which is confounded by the mice not actually developing significant MASLD pathology. I think the research question is inherently flawed as a results. How are we to ascertain if the microbiome/metabolite changes reflect early progression of MASLD or just obesity in general. If metabolic syndrome or obesity is the focus of the paper (with more endotoxemia-related data) the claims would be more accurate. As is, these mice do not exhibit liver injury at 21 weeks so I am not confident in the validity of these findings.
The microbiome analyses is very crude and limits the overall resolution of changes that may be occuring. Just looking at the F/B ratio is not enough to ascertain community diversity. I would recommend doing sequencing to correct for sampling depth and perform alpha and beta diversity metrics.
Better correlation analyses between the metabolites and gut microbial members would also strengthen claims.
As is, I strongly recommend changing the paper to remove MASLD, or conducting another experiment with a diet that is known to induce MASLD, such as a high-cholesterol, high-fat diet (1-2%).
Author Response
Comment 1: The authors make the claim that their model reflects early stages of MASH or MASLD which is confounded by the mice not actually developing significant MASLD pathology. I think the research question is inherently flawed as a result. How are we to ascertain if the microbiome/metabolite changes reflect early progression of MASLD or just obesity in general. If metabolic syndrome or obesity is the focus of the paper (with more endotoxemia-related data) the claims would be more accurate. As is, these mice do not exhibit liver injury at 21 weeks so I am not confident in the validity of these findings.
Answer: We thank the reviewer for the thoughtful comments and reflections. As the reviewer points out, in general it is very difficult to ascertain in diet models if changes reflect early progression of MASLD or just obesity in general. However, our study is a longitudinal study in which we identify changes very early in the experiment (as early as 3 weeks) and earlier than large weight gain. Most of these changes remain during the rest of the experiment and are observed at the final stage of the study, in which we can assess liver alterations by histological analysis. Although the liver of our rats was not truly inflamed (there was no MASH) nor did it have fibrosis, our results reflect histological changes (steatosis), shifts in hepatic metabolism, and a decrease in plasma bile acid (BA) content, which all reflect an early stage of MASLD. Our analysis of hepatic metabolism by intact tissue metabolomics and the fat deposition histologically observed in the liver, further support the focus of the study on early MASLD. Given the new term MASLD, we are aligning this in our article: metabolic changes or metabolic dysfunction associated with liver steatosis. In addition, as can be observed at table 1, the final body weights of the rats under HFD and CTRL diet are not that different and HFD cannot be considered obese in comparison with the control group. It is important to emphasize that we do not mention any causality in our discussion because our data cannot support it. Our focus is on a timeline of changes and alterations that happen during fat accumulation in the liver, and we think that the main claim of our study is correct. To further clarify our model and the validity of this claim, we have modified the title to “Microbiota co-metabolism alterations precede changes in the host metabolism in the early stages of diet-induced MASLD in Wistar rats” which better reflects our findings. We have also revised the text and added sentences in the discussion/ conclusions to better explain what we intended to convey.
Comment 2: The microbiome analyses is very crude and limits the overall resolution of changes that may be occuring. Just looking at the F/B ratio is not enough to ascertain community diversity. I would recommend doing sequencing to correct for sampling depth and perform alpha and beta diversity metrics.
Answer: We fully understand the concern of the reviewer because sequencing is a very common and more complete approach in the study of microbiota. We acknowledge in the manuscript that the qPCR analysis has some limitations since it does not encompass the entirety of bacterial diversity. However, qPCR is a very sensitive, reliable, and reproducible method to assess microbiota changes and alterations in specific species, and it is faster and more cost-effective, making it ideal for longitudinal studies. The primers we used were extensively validated, and the corresponding agarose gel electrophoresis was performed to validate the data. By using 7 primers, we analyze a relevant set of microbiota families and genus, going beyond the F/B ratio, which allows us to detect relevant alterations in microbiota composition, as described in tables S5 and S6 of the supplementary material. While we understand the additional insights that sequencing could provide, we think they are not needed to assess microbiota alterations in the context of a longitudinal study. Based on the literature, it has already been well established in many studies that metabolic diseases are associated with changes in bacterial diversity. Sequencing offers a comprehensive view of the entire microbial community, identifying novel species and providing higher taxonomic resolution. We chose qPCR also for translational potential. Our results were consistent across all time points adding validity to both the method used and the findings. Additionally, we have supplemented the study with longitudinal metabolomics data, and we believe these results provide sufficient support to validate our findings, without the need of more complex and expensive approaches like sequencing. Nevertheless, we have deemed it appropriate to expand the explanation of the study's limitations.
Comment 3: Better correlation analyses between the metabolites and gut microbial members would also strengthen claims.
Answer: Thank you very much for the recommendation. We strongly believe you are correct, and that Figure 6 lacked sufficient information by presenting only a heatmap. We have now added a Spearman correlation table to Figure 6 (Figure 6B) for week 3 to demonstrate that the patterns observed in the heatmap (Figure 6A) reflect actual correlations. Additionally, we have included a supplementary figure (Figure S4) comparing the data at week 21 to verify whether the correlation patterns persisted, diminished, or strengthened over time. We have also revised the text accordingly.
Comment 4: As is, I strongly recommend changing the paper to remove MASLD, or conducting another experiment with a diet that is known to induce MASLD, such as a high-cholesterol, high-fat diet (1-2%).
Answer: Although it is true that other experiments can generate a more severe phenotype, we believe we have already demonstrated, in the answer to the first comment of this reviewer, that our model is equally valid for our aims. Our goal is to study the timeline of early stages of subclinical diet-induced MASLD in Wistar rat. We think that established MASLD is intrinsically linked to many other metabolic diseases, and it is virtually impossible to use a diet model that can isolate it from the rest. A high cholesterol diet, for example, is not ideal either for studying MASLD in Wistar rats because it can induce hypercholesterolemia and other metabolic changes that may confound the specific effects of MASLD. High-fat diets more closely mimic the dietary conditions that lead to MASLD in humans, inducing obesity and fatty liver, which are frequently hand-to-hand, without the confounding effects of hypercholesterolemia. This allows for a clearer assessment of the progression of MASLD and its associated metabolic changes. Our rats under high fat diet, during the duration of the experiment, did not show hypercholesterolemia, dyslipmeia, insulin resistance or other metabolic alterations and only show a moderate weight gain with respect controls. We think that we have moderate liver alterations in an early stage of MASLD with few confusion elements from other metabolic dysfunctions. As an additional goal, the metabolites identified as altered here could indicate a shift in metabolism and serve as early warning signs that an individual might develop more severe damage if no lifestyle interventions are made. While determining severe damage and associating it with metabolites is valuable, we believe it may be even more useful to publish information on the early stages of the disease, contribute to establishing the sequence of events leading to MASLD, and, most importantly, provide insights for earlier diagnosis of the condition or other conditions associated.
Round 2
Reviewer 2 Report
Comments and Suggestions for Authors
The authors made significant efforts to address comments and improve the quality of the paper. Expanding on the limitations of these findings including: less-severe liver steatotic phenotype, depth of microbiome analysis, and conclusions made improves them overall quality of the paper. I still believe the paper should substitute "early-onset of steatosis" for MASLD, but this perspective can be argued due to the complexity of the disease and translatability between murine and clinical models.
Author Response
Comment 1: The authors made significant efforts to address comments and improve the quality of the paper. Expanding on the limitations of these findings including: less-severe liver steatotic phenotype, depth of microbiome analysis, and conclusions made improves them overall quality of the paper. I still believe the paper should substitute "early-onset of steatosis" for MASLD, but this perspective can be argued due to the complexity of the disease and translatability between murine and clinical models.
Answer: Thank you so much for your comments. We greatly appreciate that you liked our changes and that, thanks to your feedback, the article has improved significantly. Once again, we are very grateful for your second input and for helping us make our article clearer and more impactful.
We fully agree with your suggestion to reinforce the message that we are addressing an "early onset of steatosis." To address this, we have made some modifications to the text and ensured that it is clearly stated throughout the article that we are focusing on an early stage of the disease. Additionally, we have included a statement in the limitations section to emphasize that while this is an early-stage model, it does not diminish its value. Please let us know if you agree with the changes and if they are sufficiently appropriate.